Relationship between tooth macrowear and jaw morphofunctional traits in representative hypercarnivores

http://orcid.org/0000-0001-5335-4230 Tseng Z. Jack 1 2 zjt@berkeley.edu
http://orcid.org/0000-0003-1159-9154 DeSantis Larisa R. G. 3
1 Division of Paleontology, American Museum of Natural History , New York , United States
2 Department of Integrative Biology, University of California, Berkeley , California , United States
3 Department of Biological Sciences and Department of Earth and Environmental Sciences, Vanderbilt University , Nashville, Tennessee , United States
Johnson Michela
Electronic publication date: 2024 Nov 11
Publication date: 2024
Volume: 12
Electronic Location ID: e18435
Received 2024 Aug 28; Accepted 2024 Oct 10
Copyright: © 2024 Tseng and DeSantis
Copyright year: 2024
Copyright holder: Tseng and DeSantis
License: This is an open access article distributed under the terms of the Creative Commons Attribution License, which permits unrestricted use, distribution, reproduction and adaptation in any medium and for any purpose provided that it is properly attributed. For attribution, the original author(s), title, publication source (PeerJ) and either DOI or URL of the article must be cited.
License URL: https://creativecommons.org/licenses/by/4.0/

Keywords: Dentition, Carnivora, Hyaenodonta, Functional morphology, Mandible, Ecomorphology

Funding: National Science Foundation (U.S.A.) DEB-1257572 University of California Berkeley Libraries This study was supported by the National Science Foundation (U.S.A.) grant DEB-1257572 (to Z Jack Tseng). University of California Berkeley Libraries provided funds for the article processing charge. There was no additional external funding received for this study. The funders had no role in study design, data collection and analysis, decision to publish, or preparation of the manuscript.

==============================
The mammalian adult dentition is a non-renewable resource. Tooth attrition and disease must be accommodated by individuals using behavioral, physiological, and/or musculoskeletal shifts to minimize impact on masticatory performance. From a biomechanical perspective, the musculoskeletal system becomes less efficient at producing bite force for a given amount of muscle input force over an individual’s life, because tooth-food contact area increases as cusps wear. In this study we ask the question: does mandibular biomechanical performance show evidence of compensation with increasing tooth wear? We use representative taxa of three carnivoran ecomorphologies (meat specialist, scavenger, bone cracker) as a study system to compare morphofunctional data on tooth macrowear, jaw depth, bite mechanical efficiency, and jaw stress during biting. No significant shifts in adult mandibular corpus dimensions occurs in the sampled taxa as canine and carnassial teeth wear. In bone cracking spotted hyenas carnassial biting mechanical efficiency increases significantly with increasing tooth wear, with no significant change in mandibular stress. Analyses of the fossil carnivore Hyaenodon suggests an increase in canine biting efficiency with increased tooth wear, but this may reflect interspecific variation or phylogenetic contingency rather than a life history shift. Overall, these findings indicate that scavenging hyaenids and felid meat specialists do not exhibit morphofunctional compensation for the decreased mechanical capability of worn and dull teeth. Behavioral modifications, rather than musculoskeletal adjustments, may instead play a major role in maintaining food acquisition and processing capabilities for individuals surviving into advanced ontogenetic age and tooth wear. These observations highlight the mammalian masticatory system as having a dynamic performance profile through its useful lifespan, and encourage a more nuanced understanding of past and present carnivore guilds by considering wear-dependent performance changes as a possible source of selection.

Introduction

Diphyodonty, the condition of having two generations of teeth throughout an individual’s life, is a salient feature of crown mammals (Luo, Kielan-Jaworoska & Cifelli, 2004). Evolutionary benefits of having a permanent or adult set of dentitions may include functional consistency and stability in support of heterodonty, maintenance of precise occlusal performance, and reduction of energetic budget spent on dental growth. However, a principal trade-off of diphyodonty is the constraint of the permanent dentition as a non-renewable tissue. Wear or breakage to the adult teeth may affect their function, and any performance compensation in response must be made from other parts of the masticatory system because tooth enamel cannot repair itself. The evolutionary manifestation of this key property of mammalian dental tissues can be observed in species ranging from shrews to elephants, in which tooth wear severity is a limiting factor in individual lifespans (Lucas & van Casteren, 2014).

A concomitant evolutionary innovation alongside a diphyodont and heterodont dentition in mammals is a many-to-one form-function linkage of their lower jaws. The post-K-Pg radiation of mammalian taxonomic diversity also reflects a radiation of jaw shape disparity (Tseng et al., 2023). However, jaw biomechanical performance, specifically stiffness, is both elevated and less variable in crown mammals than in other vertebrates. This suggests that a stiff jaw is a synapomorphic condition of crown mammals regardless of feeding ecology.

The combination of a stiff lower jaw bone and a diphyodont, heterodont dentition underlies the diversity of feeding ecologies observed across living mammals (Jones et al., 2009). Although tooth wear and its corresponding functional changes is a fact of life for most mammals, it is unclear whether the universally stiff jaws of mammals compared to other vertebrates implies that overall biting biomechanical performance is maintained across mammalian tooth wear stages. The amount of pressure or stress that can be generated at the tooth-food contact surface is inversely proportional to the area of that contact; for a given amount of force generated, stress is equal to that force divided by the area through which the force is applied. For resistant food items that require crushing, cracking, or shearing, the most efficient way to generate a fracture in the food bolus is to concentrate the bite force over a small occlusal area of the tooth crown. As teeth wear, the occlusal area enlarges, and thus the same masticatory task would require higher forces to generate the same pressure/stress at the tooth-food interface. Again, because tooth enamel wear is irreversible, any compensation to biting performance must come from other aspects of the masticatory system.

In this study, we ask whether the lower jaw exhibits different morphofunctional characteristics according to the severity of tooth wear. We also ask whether any such morphofunctional traits support the identification of convergent feeding ecologies in the fossil record. We use a carnivoran study system, well-known for its strong link between tooth wear and feeding ecology (DeSantis et al., 2015; DeSantis, 2016; Burtt & DeSantis, 2022), to test two hypotheses:

H1: Bone cracking and scavenging ecological morphologies (ecomorphs; Werdelin & Gittleman, 1996; Van Valkenburgh, 2007), represented among living carnivorans by large hyaenids with mechanically demanding diets, exhibit morphofunctional compensation of decreased force to area ratio for a given input muscle force as tooth wear increases. There should be a significant difference in mechanical efficiency, strain energy, and/or jaw dimensions across tooth macrowear categories. By contrast, meat specialists (represented in this study by large felids), which do not experience high mechanical demands, do not exhibit morphofunctional compensation for tooth wear.

H2: The fossil taxon Hyaenodon, long interpreted as an ecological avatar of extant large-bodied hyenas, should exhibit similar relationships between tooth macrowear and morphofunctional trait variation as extant bone cracking and scavenging ecomorphs represented by some hyenas. Such similarity reflects similar ecomorphological adaptation between Hyaenodon and extant hyaenids. It is worth noting that a Miocene hyaenid adaptive radiation produced a diversity of jackal-like and wolf-like forms, and living hyaenids include an ant-specialist (Werdelin & Solounias, 1991; Galiano et al., 2022); for the purpose of this study, we focused our comparisons on the three bone cracking and scavenging hyaenids genera Crocuta, Hyaena, and Parahyaena.

Materials and Methods

All morphofunctional data analyzed in this study are based on 2D photographs of hemimandible specimens in two museum collections: the American Museum of Natural History (AMNH) and the University of Michigan Museum of Zoology (UMMZ). A total of 54 specimens representing six genera were included in the analyses (Table 1). Each AMNH specimen was placed onto the scanning area of a Dell AIO A960 Flatbed Scanner in its natural resting position with the lateral side facing the scanning bed. A metric scale bar was placed next to the specimen. A color image at a resolution 600 dpi was then captured and saved as a tiff image file. UMMZ specimen images were downloaded from the Animal Diversity Web (https://animaldiversity.org) under an CC BY-NC-SA 3.0 license by P. Myers.

Table 1 Sample size and feeding ecology assignments of taxa examined in this study.

Ecomorph	Genus	Species	Sample size	
Bone cracker	Crocuta	crocuta	10	
Meat specialist	Acinonyx	jubatus	10	
	Panthera	leo	11	
		Meat specialist total	21	
Scavenger	Hyaena	hyaena	7	
	Parahyaena	brunnea	7	
		Scavenger total	14	
Fossil	Hyaenodon	paucidens	1	
	Hyaenodon	exiguus	1	
	Hyaenodon	crucians	1	
	Hyaenodon	luskensis	1	
	Hyaenodon	brevirostris	1	
	Hyaenodon	horridus	1	
	Hyaenodon	cruentus	3	
		Fossil total	9	
		All models total	54	

Tooth macrowear analysis

We categorized wear stages of all canine and carnassial teeth in the dataset using the scheme defined in DeSantis et al. (2017). Each tooth was given a score from 1 to 3, where a score of 1 indicates little to no occlusal wear with little or no dentine exposed, two indicates moderate occlusal wear with dentine exposure, and three indicates extensive occlusal wear with dentine exposure larger in area than the remaining enamel at the wear surface.

Jaw measurements

We used FIJI (Goldstein et al., 2018) to take all linear measurements. Each image was opened in FIJI, calibrated by setting the scale according to the length of 10 mm on the scale bar included in each photograph, and then using the line tool to make measurements. Jaw length measurements were taken on all specimens by taking the distance between the anterior boundary of the first lower incisor and the mandibular bone, and the posterior-most point on the condylar process. Two additional measurements were taken as proxies for the bending strength of the mandibular ramus below canine and carnassial bite positions, respectively: depth of ramus at the post-carnassial position, and depth of ramus at the post-canine position. Lastly, we record total jaw model volume from the finite element models of each specimen, the construction of which is detailed below.

Biomechanical performance estimates (Fig. 1)

Each specimen image was converted into a high contrast image that represents the jaw in black pixels and surrounding space in white pixels. We used the magnetic lasso tool in GIMP 2.10.20 to select the jaw, reversed the object selection, and removed background pixels. The high contrast image was then exported as PNG files and next converted into an outline bound by nodes within Inkscape version 0.48. The outlines were saved as .dxf files. Next, the outline shape was extruded with an arbitrary thickness of height 10 using OpenSCAD version 2014.01.29 and converted into a mesh file in STL format. The extruded shape was then improved for triangular element count, aspect ratio, and evenness in Geomagic Wrap 2019. The imported stl meshes were first refined to represent at least 60k triangular faces, then cleaned using the ‘quick smooth’ tool. The meshes were then decimated to a target triangle face count of 50k, with triangle face dimensional aspect ratio constrained to 10 or less. Lastly, the meshes were subjected to the mesh improvement tool ‘mesh doctor’ and then alternated with mesh decimation until the mesh improvement tool no longer detected any mesh issues. The final clean meshes were then exported as stl files and used for 2D finite element modeling.

Figure 1 Examples of bite simulation models for each of the three feeding ecologies and the fossil taxon studied.

(A) Acinonyx jubatus, a meat specialist; (B) Hyaena hyaena, a scavenger; (C) Panthera leo, a meat specialist; (D) Parahyaena brunnea, a scavenger; (E) Crocuta crocuta, a bone cracker; (F) Hyaenodon crucians, a fossil carnivore. Green silhouettes represent cranial reference shapes. Muscle insertion areas: temporalis (red), zygomaticomandibularis/deep masseter (blue), masseter/superficial masseter (orange). Centroid points for guiding muscle vector orientations are shown in the same colors as their respective muscle groups. Abbreviations: M, masseter; Mc, masseter centroid; T, temporalis; Tc, temporalis centroid; ZM, zygomaticomandibularis; ZMc, zygomaticomandibularis centroid.

We used Strand7 finite element analysis software version 2.4.6 to estimate biomechanical performance traits from the extruded mandibular meshes. Meshes were checked and cleaned using the automatic clean mesh tool in Strand7. If errors were detected during this mesh cleaning step, the mesh was taken through the improvement procedure outlined in the previous paragraph and reimported into Strand7 until no errors were detected. The mesh file was then exported once again as stl files for muscle and tooth enamel mesh generation.

The vetted mesh file form Strand7 was then reimported into Geomagic Wrap to generate muscle and tooth enamel mesh groups. Three muscle groups were delineated on the ascending ramus of the jaw shape based on previous descriptions of musculoskeletal anatomy in spotted hyenas (Tseng & Binder, 2010) and carnivorans in general (Evans & Christensen, 1979; Tseng & Stynder, 2011). The temporalis, superficial masseter, and deep masseter muscles were included in the biting simulation models; given the 2D approach, muscles that largely contribute to lateral jaw movements such as pterygoideus muscles were not modeled. The enamel crown of the canine and cheek dentitions on the mesh models were highlighted based on the enamel crown areas visible from specimen photographs. The highlighted triangle faces were then copied and pasted as a separate mesh group to allow different material properties to be defined during the model simulation step (see below).

Photographs of cranial specimens for all six genera included in the analyses were used to generate reference cranial meshes using the same protocol described above for mandible mesh generation. The reference cranial meshes (one for each genus) were then imported and scaled to each mandible mesh for mandibular muscle force contraction vector estimation. We scaled the cranial reference to each mandible mesh by aligning the dorsal face of the mandible condyle with the ventral face of the mandibular fossa on the temporal bone, and the distal face of the lower canine tooth to the mesial face of the upper canine tooth, respectively. The cranial reference mesh was then rotated away from the mandible mesh by 30 degrees, representing an average gape for carnivorans (Bourke et al., 2008). Next, centroid points were generated for each of the three muscle groups. Muscle origination areas were highlighted on the cranial reference mesh, extruded to a thickness of 1 mm, and a ‘center of mass’ point was calculated using the function of the same name in Geomagic Wrap. These 3D centroid points were used as a reference to create 2D centroid coordinates directly on the surface of the original 2D muscle highlights. The x and y values of the centroid coordinates were recorded for each jaw-cranial mesh combination.

Next, muscle forces, joint and bite point constraints, and material properties were defined to fully parameterize the jaw model. The amount of force generated by each muscle insertion area (towards the centroid points on cranial reference meshes) was set to be proportional to the surface area represented by the muscle insertion meshed, multiplied by 0.3 N based on a maximum muscle contraction force of 0.3 N/mm2 (Wroe, McHenry & Thomason, 2005). We used muscle insertion area as a proxy for muscle contractile force rather than estimated physiological cross section area because 3D information is not available from the 2D specimen photographs. It is important to note that the underlying assumption of our approach is that muscle insertion area is a good approximation of its force production capability. We argue that this is a reasonable assumption, as it standardizes our interspecific comparisons of biomechanical response to biting scenarios as a product of overall muscle contraction rather than species-specific muscle activation ratios, for which no empirical data are available.

We used the BoneLoad program (Grosse et al., 2007) to generate distributed force vectors over muscle insertion areas to mimic muscle contraction. The force loaded meshes were then reimported into Strand7, where free body movement constraints and material properties were defined. Although all parts of the jaw model are represented by 2D plate elements, we defined a thickness of 10% of the maximum model length to enable calculation of in-plane bending stress. A negligible thickness of 0.0001 mm was assigned to muscle attachment meshes to simulate the direct action of muscle fibers pulling on the underlying bone. Young’s (elastic) modulus of 20 GPa (gigapascals) and Poisson ratio of 0.3 were assigned to the bone and muscle portions of the mesh model. The tooth enamel portion of the model was assigned a modulus of 80 GPa and Poisson ratio of 0.3.

Three different bite scenarios were simulated: canine bite, canine pull, and carnassial (m1 in carnivorans, m3 in Hyaenodon) bite (Fig. 1). In all three cases we placed a full nodal constraint at the center of the condylar process that prevented any translational or rotational movement. In the canine bite scenario, a partial nodal constraint was placed at the tip of the canine tooth to prevent dorsoventral movement but allowing anteroposterior movement. This scenario simulated full muscle contraction during jaw closure and food contact at the tip of the canine. In the canine pull scenario, an anteriorly directed force equivalent to 10% of total muscle input force was placed at the same canine constraint as in the canine bite scenario, and all other conditions are identical to the canine bite scenario. This scenario simulated full muscle contraction during jaw closure, with a bite point at the canine and an external force from struggling prey. Lastly, in the carnassial bite scenario, the jaw joint constraint is as in the other two scenarios, but a cusp nodal constraint is placed at the carnassial paraconid instead of the canine tooth. This scenario simulated jaw closure with full muscle contraction during mastication at the carnassial tooth.

All three bite scenarios were solved using Strand7’s linear static solver function. We then extracted both qualitative and quantitative data from the three bite scenarios. Output nodal reaction forces at the tooth cusp constraints were measured for the canine and carnassial bite scenarios and divided by total input muscle force to derive mechanical efficiency. Stored strain energy (in Joules), a measure of the work done by an input load in deforming a structure under load given a set of constraint conditions, was measured for each of the three scenarios. Lastly, heatmap visualizations of von Mises stress, which summarizes the distribution of forces on a structure under load, were generated from one model for each of the extant genera, and for all fossil specimens modeled.

Statistical analyses

We evaluated data support for our stated hypotheses using analysis of variance (ANOVA). The tooth macrowear categories were used as groups, and ANOVA tests were conducted separately for the canine and carnassial macrowear of a bone cracking hyaenid (Crocuta crocuta), two scavenging hyaenids (Hyaena hyaena and Parahyaena brunnea), two large meat specialist felids (Panthera leo and Acinonyx jubatus), and the fossil genus Hyaenodon. The five extant taxa were chosen as representative extant species in their respective ecomorphs that have been used in comparisons to, and in ecomorphological reconstructions of, fossil carnivores (Werdelin & Gittleman, 1996; Van Valkenburgh, 2007). We note that other, unsampled extant taxa of similar ecomorphs may not necessarily exhibit similar tooth macrowear to jaw mechanics relationships exhibited by the sampled taxa; thus, functional or evolutionary pattern extrapolations beyond the taxonomic sampling covered in this study should be done with caution. Morphofunctional traits evaluated against macrowear categories included input muscle force (in Newtons), output bite point reaction force (in Newtons), mechanical efficiency (output bite point reaction force/input muscle force), strain energy (J), total model volume (mm3), jaw length (mm), and jaw width (mm). Additionally, results were visualized as boxplots using R programming packages ggplot2 and ggpubr (Wickham, 2016). All statistical tests were conducted in R using the aov function in the core R library.

Results

Tooth macrowear analysis

All but one meat specialist carnassial examined (20 out of 21) exhibited little to no macrowear. By contrast, all three categories of macrowear are recorded for the canine position of meat specialists (Table S1). The majority of canine and carnassial macrowear scores are 2 or 3 in the scavenger data partition, and in bone crackers about half of the specimens have macrowear scores of 2 or 3. The majority of Hyaenodon specimens have a macrowear category of 2 or 3 in both canine and carnassial tooth positions.

Jaw measurements

Meat specialists in our dataset have a mean jaw length of 167.06 mm, mean jaw depth at canine of 31.60 mm, and mean jaw depth at m1 of 31.46 mm. Scavengers have a mean jaw length of 165.24 mm, canine jaw depth of 32.58 mm, and m1 jaw depth of 37.72 mm. Bone crackers have a mean jaw length of 162.97 mm, canine jaw depth of 29.88 mm, and carnassial jaw depth of 39.13 mm. Lastly, the Hyaenodon specimens studied have a mean jaw length of 179.78 mm, canine jaw depth of 24.96 mm, and carnassial jaw depth of 35.40 mm. Based on these measurements, meat specialists have a nearly 1:1 ratio of jaw depth at the carnassial vs. the canine, scavengers have 16% deeper mandibular ramus at the carnassial compared to the canine position, and bone crackers have ~30% deeper ramus at the carnassial compared to the canine position. In this regard, Hyaenodon is closest to bone crackers in having 41.6% deeper jaws at the carnassial compared to the canine position.

No clear patterns of jaw measurement differences across macrowear categories are present for either the canine or carnassial data of all ecomorph partitions (Fig. 2). Furthermore, none of the ANOVA tests returned statistically significant results (p values range from 0.90 to 0.06; Table 2).

Figure 2 Boxplots of jaw dimension values by macrowear category.

(A) Canine macrowear vs. jaw model volume. (B) Carnassial macrowear vs. jaw model volume. (C) Canine macrowear vs. jaw length. (D) Carnassial macrowear vs. jaw length. (E) Canine macrowear vs. jaw depth at post-canine position. (F) Carnassial macrowear vs. jaw depth at post-carnassial position.

Table 2 Results of ANOVA tests of morphofunctional traits across tooth macrowear categories by feeding ecology.

		Bone cracker	Meat specialist	Scavenger	Hyaenodon	
Macrowear category	Morphofunctional trait	F	p	F	p	F	p	F	p	
Canine macrowear	Input muscle force (N)	4.52	0.07	0.71	0.41	0.33	0.58	0.07	0.80	
	Mechanical efficiency	3.27	0.11	1.27	0.28	3.01	0.11	8.95	0.03	
	Strain energy (Bite; J)	0.02	0.90	0.48	0.50	0.20	0.66	1.43	0.29	
	Strain energy (Pull; J)	0.14	0.72	0.05	0.82	0.19	0.67	1.44	0.28	
	Model volume (mm3)	0.11	0.75	0.10	0.76	0.49	0.50	0.57	0.48	
	Jaw length (mm)	1.75	0.22	0.05	0.83	0.44	0.52	0.55	0.49	
	Jaw depth (mm)	3.14	0.11	0.02	0.90	1.94	0.19	0.06	0.81	
Carnassial macrowear	Input muscle force (N)	4.04	0.08	0.01	0.92	0.61	0.45	0.10	0.77	
	Output muscle force (N)	6.32	0.04	0.06	0.82	0.83	0.38	0.16	0.70	
	Mechanical efficiency	9.31	0.02	1.31	0.27	3.26	0.10	0.01	0.91	
	Strain energy (Bite; J)	3.81	0.09	0.33	0.57	0.03	0.87	2.17	0.18	
	Model volume (mm3)	0.04	0.86	0.06	0.81	0.77	0.40	0.19	0.68	
	Jaw length (mm)	1.45	0.26	1.12	0.30	2.12	0.17	0.43	0.53	
	Jaw depth (mm)	4.76	0.06	1.13	0.30	4.02	0.07	0.39	0.55	
Note:

p values < 0.05 are indicated in bold font.

Biomechanical performance estimates

Bone crackers at later macrowear stages tend to possess larger muscle insertion areas and therefore larger muscle input forces than other feeding ecologies, even though they are not overall the largest individuals in the dataset (Fig. 3). Canine bite mechanical efficiency values do not exhibit clear trends across macrowear categories in any feeding ecologies; however, bone crackers show increasing carnassial bite mechanical efficiency with increasing macrowear (Fig. 3D, Table 2; F = 9.31, p = 0.02). Hyaenodon exhibit increasing canine mechanical efficiency (F = 8.95, p = 0.03; Table 2) but no change in carnassial mechanical efficiency with increasing macrowear.

Figure 3 Boxplots of input muscle force and output mechanical efficiency values by macrowear category.

(A) Canine macrowear vs. input muscle force. (B) Carnassial (m1) macrowear vs. input muscle force. (C) Canine macrowear vs. canine bite mechanical efficiency. (D) Carnassial macrowear vs. carnassial bite mechanical efficiency.

Meat specialists tend to exhibit increased strain energy (lower work efficiency or stiffness) at macrowear category 3 compared to other categories in canine biting (Figs. 4A, 4B), and a larger spread of strain energy values at macrowear category 1 in carnassial bite simulations (Fig. 4C). None of the strain energy patterns are statistically significant (Table 2). In m1 bite reaction force, bone crackers alone exhibit a significant increase with increasing macrowear (F = 6.32, p = 0.04; Table 2, Fig. 4D), mirroring the pattern observed in m1 mechanical efficiency (Fig. 3D).

Figure 4 Boxplots of biomechanical model strain energy and bite point reaction force values by macrowear category.

(A) Canine macrowear vs. canine bite strain energy. (B) Canine macrowear vs. canine pull strain energy. (C) Carnassial (m1) macrowear vs. m1 bite strain energy. (D) Carnassial macrowear vs. m1 output bite point reaction force.

Heatmap visualization of von Mises stress in exemplary jaw models shows qualitatively that meat specialists tend to experience higher stresses than other feeding ecologies. In all canine bite simulations, the largest region of elevated stress is in the transition between the horizontal and ascending rami, immediately posterior to the carnassial (Figs. 5A–5E). Bone crackers exhibit the lowest stress in the core of the horizontal ramus compared to other ecomorphs, displaying parallel strips of elevated stress at the dorsal and ventral edges of the mandible, respectively. All Hyaenodon specimens studied show a similar strip of low stress region along the length of the horizontal ramus in patterns most similar to bone crackers (Figs. 5F–5N). In canine pull simulations the overall von Mises stress distributions are similar to those observed in canine bite simulations. The major difference is a relatively more stressed vender border along the horizontal ramus when a canine bite is combined with an anterior pulling force (Fig. 6).

Figure 5 Heatmap visualization of von Mises stress from canine bite simulations.

(A) Acinonyx jubatis, AMNH-VP (extant element collection of AMNH Department of Vertebrate Paleontology) 2502; (B) Panthera leo, UMMZ 114804; (C) Crocuta crocuta, UMMZ 114799; (D) Hyaena hyaena, AMNH-VP 1544; (E) Parahyaena brunnea, UMMZ 95748; (F) Hyaenodon brevirostris, F:AM (Frick collection of the AMNH) 75629; (G) H. crucians, F:AM 75596; (H) H. cruentus, F:AM 75607; (I) H. cruentus, F:AM 75692; (J) H. cruentus, F:AM 75729; (K) H. exiguus, AMNH 55314; (L) H. horridus, F:AM 75704; (M) H. luskensis, F:AM 75606; (N) H. paucidens, AMNH 647.

Figure 6 Heatmap visualization of von Mises stress from canine pull simulations.

(A) Acinonyx jubatis, AMNH-VP (extant element collection of AMNH Department of Vertebrate Paleontology) 2502; (B) Panthera leo, UMMZ 114804; (C) Crocuta crocuta, UMMZ 114799; (D) Hyaena hyaena, AMNH-VP 1544; (E) Parahyaena brunnea, UMMZ 95748; (F) Hyaenodon brevirostris, F:AM (Frick collection of the AMNH) 75629; (G) H. crucians, F:AM 75596; (H) H. cruentus, F:AM 75607; (I) H. cruentus, F:AM 75692; (J) H. cruentus, F:AM 75729; (K) H. exiguus, AMNH 55314; (L) H. horridus, F:AM 75704; (M) H. luskensis, F:AM 75606; (N) H. paucidens, AMNH 647.

The carnassial bite simulations differ from canine simulations in having more limited regions of high stress (Fig. 7). Meat specialists and scavengers tend to exhibit a continuous path of elevated stress connecting the dorsal and ventral horizontal stress paths ventral to the carnassial bite position. The bone cracking Crocuta and the morphologically robust scavenger Parahyaena show the least amount of elevated stress along that dorsoventral path. Similarly, the von Mises stress patterns for carnassial biting in Hyaenodon specimens tend to show two separate elevated stress paths at the dorsal and ventral margins of the ramus, respectively. As expected, the unloaded region of the mandible anterior to the bite point does not show elevated von Mises stress in any of the models visualized.

Figure 7 Heatmap visualization of von Mises stress from carnassial bite simulations.

(A) Acinonyx jubatis, AMNH-VP (extant element collection of AMNH Department of Vertebrate Paleontology) 2502; (B) Panthera leo, UMMZ 114804; (C) Crocuta crocuta, UMMZ 114799; (D) Hyaena hyaena, AMNH-VP 1544; (E) Parahyaena brunnea, UMMZ 95748; (F) Hyaenodon brevirostris, F:AM (Frick collection of the AMNH) 75629; (G) H. crucians, F:AM 75596; (H) H. cruentus, F:AM 75607; (I) H. cruentus, F:AM 75692; (J) H. cruentus, F:AM 75729; (K) H. exiguus, AMNH 55314; (L) H. horridus, F:AM 75704; (M) H. luskensis, F:AM 75606; (N) H. paucidens, AMNH 647.

Discussion

Bite simulation and macrowear analyses of hypercarnivore mandible models show that for felid meat specialists and hyaenid scavengers there is no evidence of morphofunctional compensation in mandibular performance with increased tooth wear. However, there is a statistically significant increase in carnassial bite mechanical efficiency with increasing macrowear in bone cracking spotted hyenas. There is no correlation between macrowear and either jaw strain energy (a measure of work efficiency or stiffness) or jaw dimensional changes in any of the feeding ecologies studied. These results provide only partial support for our first hypothesis (H1), that bone crackers and scavengers exhibit morphofunctional compensation in mandible performance with increasing tooth macrowear whereas meat specialists do not.

The extinct carnivore Hyaenodon shared no statistically significant wear-dependent morphofunctional shifts with any of the extant feeding ecologies. Instead, the fossil taxon exhibits increased canine bite mechanical efficiency with increased tooth macrowear, differing from the bone crackers which show carnassial mechanical efficiency increase with macrowear. These findings provide no biomechanical support for the prior interpretation of Hyaenodon (as the name also suggests) as ecological equivalents of hyaenids in their respective paleoguilds. Therefore, we reject our second hypothesis (H2), that Hyaenodon and extant bone cracking and scavenging hyaenids share similar patterns of morphofunctional compensation with increasing tooth macrowear.

Previous studies on the functional morphology of Hyaenodon suggest a semi-arboreal locomotor ecology for H. exiguus (Pfaff et al., 2017), with comparable or more specialized dental crown features than the most specialized feliforms (Lang, Engler & Martin, 2022), reduced zygomatic arch robustness associated with capability for higher gape (De Iuliis, 1993), and similarity to hyenas or lions in dental microwear depending on geographic region (Bastl, Semprebon & Nagel, 2012). What emerges from theses studies and new findings reported in the current study is that (1) none of the sampled taxa (large felids and bone cracking and scavenging hyaenids) converge on Hyaenodon in terms of the morphofunctional traits analyzed, and (2) there is diversity in the range of dietary ecologies within the genus Hyaenodon. Therefore, the lack of a match in the morphofunctional traits measured in this study between Hyaenodon and extant hypercarnivore feeding ecologies may reflect a combination of unique niches occupied by Hyaenodon, a possible mixture of ecomorphs represented in our Hyaenodon dataset, or more generally a fundamental phylogenetic difference in form-function relationships in the extant carnivorans sampled and the extinct hyaenodontid lineage represented by Hyaenodon. We combined different species of Hyaenodon into a single dataset because of the small fossil sample sizes available; this may have reduced the functional morphological signal available in the data by mixing multiple ecomorphs. Future research that focuses on larger single-taxon samples of Hyaenodon will permit a test of this interpretation.

The absence of morphofunctional correlates of tooth macrowear in the meat specialist and scavenging hypercarnivore species studied indicates that either (1) tooth wear has no significant impact on biting performance, or (2) tooth wear does influence biting performance but there is no morphofunctional compensation. In the case of meat specialists, it may be that advanced tooth macrowear is rarer than in other ecomorphs with more mechanically demanding diets (Table S1), rendering morphofunctional compensation unnecessary or too subtle to be detected with the current dataset. Behavioral and natural history observations from living felids (which are collectively categorized as meat specialists) offer possible explanations for the observed macrowear patterns in the large felids studied. In some extant puma populations, both age-dependent and life stage-dependent differences in predation patterns have been observed (Elbroch, Feltner & Quigley, 2017; Elbroch & Quigley, 2019). Dispersing pumas tend to go after smaller prey, whereas older pumas tend to take down larger prey. Both observations suggest that behavioral shifts play a role in predation and constitute another dimension of compensation for individual condition and age (which includes tooth wear) beyond morphofunctional traits. The presence of other predators, including the relative abundance of co-occurring wolves in North America, can also mediate dietary choices including the size and condition of prey species in pumas (Kortello, Hurd & Murray, 2007; Bartnick et al., 2013)—with pumas consuming prey with the greatest range of body sizes as compared to neotropical carnivores (Cruz et al., 2022). On the other hand, no significant dietary differences were observed among individuals of a high-density jaguar (Panthera onca) population (Foster & Harmsen, 2022). Thus, there may be a large range of behavioral plasticity that masks any morphofunctional response to decreased masticatory capability with increased tooth macrowear, at least in meat specialists. More generally, the potential presence of interactions and trade-offs between feeding and hunting (prey handling) strategies at the macroevolutionary scale may impose bounds on how tooth macrowear and jaw mechanics can vary (for example, the craniodental complex in sabertooths; Slater & Van Valkenburgh, 2008; Chatar et al., 2024). The potential patterns and mechanisms underlying these complex interactions require a comprehensive examination of macrowear and jaw mechanics across a broader phylogenetic sample of meat specialists that is beyond the scope of the current study.

In the bone cracking spotted hyenas (Crocuta crocuta), there is a range of hunting group sizes that correlates with individual age. Older spotted hyenas tend to hunt alone more frequently than younger individuals (Holekamp et al., 1997). In contrast, the scavengers brown hyenas (Parahyaena brunnea) and striped hyenas (Hyaena hyaena) tend to hunt and scavenge solo regardless of age, and instead scavenge in larger groups where all individuals access a similar food source (Owens & Owens, 1978; Watts & Holekamp, 2007). Our findings are consistent with these observed behavioral differences. Bone crackers (as represented by the spotted hyena, the only living taxon categorized as such a specialist by Werdelin & Gittleman, 1996) exhibit significantly increased carnassial bite mechanical efficiency with tooth macrowear typical of older individuals who hunt alone more frequently. Scavenging hyaenids show a non-significant increase in mechanical efficiency from the lowest macrowear category to the higher categories (Fig. 3D), and correspondingly do not show age-related differences in hunting strategy. Furthermore, the absence of wear-dependent morphofunctional changes in other measured traits (jaw dimensions, canine and carnassial bite strain energy, canine bite mechanical efficiency) in living bone crackers may be in part explained by social rank structured feeding behavior in that species; higher ranked individuals have preferential access to food resources regardless of whether those same individuals were responsible for the acquisition of a particular meal (Kruuk, 1972). Priority access to softer parts of a prey carcass by virtue of high social rank would permit some individuals to obtain high quality food even if their masticatory system performs suboptimally for mechanically demanding tasks because of tooth wear and damage.

One strength of our 2D based approach to estimating biomechanical performance is the ability to incorporate larger sample sizes in our finite element modeling compared to most previous studies of similar scope. The current paradigm of using FEA to correlate organismal form and function often relies on only one or two specimen models per species because of the time-consuming nature of FE protocols (Tseng, 2021). As such, no previous studies have examined individual differences in the biomechanical traits analyzed herein as a consequence of tooth wear and tear.

On the other hand, the 2D modeling approach limits our examination of biomechanical performance to the dorsoventral plane. The origin of mammalian mastication/chewing has been speculated to involve pitch and yaw components that provide more nuanced movements of the hemimandibles and thus angles of occlusion (Bhullar et al., 2019). Although the plane of wear on the carnassial teeth of carnivorans and hyaenodontids are largely in the dorsoventral direction, indicating the principal movement of occlusion to be dorsoventrally oriented, there may be important shear forces on the masticatory system that results in this study could not account for. Future studies of form-function linkage in a tooth macrowear context would benefit from a critical analysis of the extent that 3D information is consistent with, or adds substantially to, the 2D biomechanical data collected in the present study.

Conclusions

In this study we hypothesized that bone cracking and scavenging hypercarnivores should exhibit morphofunctional compensation with more severe tooth macrowear, whereas meat specialists do not have the mechanical need to make such adjustments. We found only partial support for this prediction, with results showing that the carnassial bite mechanical efficiency of bone cracking ecomorphs is the only performance attribute that is significantly correlated with the extent of tooth macrowear. We further hypothesized that the extinct carnivore Hyaenodon, commonly thought to be functionally convergent with extant hyaenids, would share similar patterns of morphofunctional compensation with tooth wear. We found that Hyaenodon is unique among the hypercarnivores studied in exhibiting canine bite mechanical efficiency increase with tooth macrowear. The incorporation of tooth macrowear patterns into assessments of morphofunctional traits provides an explicit link between form-function linkages at the interspecific level, and tooth wear and age at the individual level. These findings support the inference that rather than treating feeding ecologies as static and characterizable by single specimen models, the morphofunctional trajectories of tooth use, tooth wear, and jaw mechanics can provide an added dimension of biomechanical performance profiling for a given taxon. These observations highlight the mammalian masticatory system as having a dynamic performance profile through its useful lifespan, and encourage a more nuanced understanding of past and present carnivore guilds by considering wear-dependent performance changes as a possible source of selection.

Supplemental Information

Supplemental Information 1 Raw morphofunctional data for all specimen models used in this study.

Raw data containing model inputs/outputs from extruded 2D jaw models subjected to linear static finite element simulations. Ecomorph, carnivore category assigned to taxon. Finput_N, total input muscle force. Foutput_C_N, output node reaction force at canine tooth. Mec, mechanical efficiency (Foutput:Finput) of canine bite simulation. Foutput_m1_N, output node reaction force at carnassial toth (lower first molar). MEm1, mechanical efficiency of carnassial bite simulation. Sec_J, stored strain energy value from canine bite simulation. SEm1_J, stored strain energy value from carnassial bite simulation. SEcpull_J, stored strain energy value from canine pull simulation. Volume_mm3, jaw model volume. L_mm, jaw model length. Dm1_mm, jaw model depth at carnassial tooth position. Dc_mm, jaw model depth at canine position. macrowear_c, macrowear score for canine tooth. macrowear_m1, macrowear score for carnassial tooth.

We acknowledge N. Lo and L. Chen for their assistance with image curation and model preparation. E. Westwig (formerly of AMNH) assisted with access to the AMNH Mammalogy collections. J. Flynn and R. O’Leary assisted with access to the AMNH Paleontology collections. ZJT thanks J. Liu for discussion during the initial stages of manuscript writing. Editor M. Johnson, reviewer D. Tamagnini, and an anonymous reviewer provided supportive and constructive feedback that improved the overall scope and flow of the manuscript.

Additional Information and Declarations

Competing Interests

Author Contributions

Data Availability

The authors declare that they have no competing interests.

Z Jack Tseng conceived and designed the experiments, performed the experiments, analyzed the data, prepared figures and/or tables, authored or reviewed drafts of the article, and approved the final draft.

Larisa R. G. DeSantis conceived and designed the experiments, performed the experiments, analyzed the data, prepared figures and/or tables, authored or reviewed drafts of the article, and approved the final draft.

The following information was supplied regarding data availability:

All raw measurements and simulation outputs are available in the Supplemental File.

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
