# Peer review of "Relationship between tooth macrowear and jaw morphofunctional traits in representative hypercarnivores"

_PeerJ, doi:10.7717/peerj.18435_

## Round 0.1 · original submission · Minor Revisions

Both reviewers believe the manuscript to be promising and only have minor comments. However, please acknowledge the concern of Reviewer 2 about the limited number of taxa for each ecomorph and consider an addition of potential taxa.

·

Basic reporting

I revisioned the manuscript entitled "Relationship between tooth macrowear and jaw morphofunctional traits in hypercarnivores", which aims to clarify the role of macrowear on the dentition of multiple living and extinct carnivorans by clarifying its impact as an evolutionary driver.

The manuscript seems really promising in its current state. In particular, the analytical framework appears to be innovative and the emerging results are extremely useful for research on carnivoran biology. The writing is overall really good, except for some rare typos, mainly affecting the Abstract/Introduction sections (see my detailed comments in the attached .pdf file).

My few suggestions are mainly focused on two aspects, that are: the importance of introducing slightly better the role of fossil diversity and disparity occurred in extinct hyenas and the potential improvement of a paragraph in the Discussion by elaborating on the importance, concerning active predators, of the tradeoff between feeding biomechanics and the ability to resist unpredictable loadings deriving from prey handling.

I think the manuscript in its current state is almost ready for publication and for all the abovementioned reasons I suggest a 'Minor Revision' outcome. As mentioned above, I attach a .pdf file including all my comments in detail.

Davide Tamagnini

Experimental design

Even if I am not an expert in biomechanical studies, the adopted experimental design seems really adequate to me.

Validity of the findings

no comment

Reviewer 2 ·

Basic reporting

This manuscript investigates whether increasing tooth wear, which can reduce bite force, is associated with mandibular morphofunctional compensation in extant hypercarnivores and fossil Hyaenodon.
This study shows an increase in carnassial biting mechanical efficiency with increasing tooth wear in bone crackers, and no evidence of compensation in scavengers or meat specialists. It also shows an increase in canine biting efficiency with increased tooth wear in Hyaenodon. The authors argue that their findings do not support ecological similarities between Hyaenodon and extant hyaenids.

I find the topic addressed in this study interesting. My major concern is the choice of a limited number of taxa to represent the as three ecomorphs.
Extant hypercarnivores are represented by a larger number of species across several families within the Carnivora, although this study sampled only five species from two families. The “bone cracker” is represented by one species, the spotted hyena; the “meat specialist” by two felid species; and the “scavenger” by two hyaenid species. I am concerned that the results derived from these species alone are sufficient to generalize for the ecomorphs. It might be more appropriate to refer them as “bone cracking hyaenids”, “large felids”, and “scavenging hyaenids”. Alternatively, the unit of comparison might be the five species (spotted hyena, lion, cheetah, brown hyena, and striped hyena) rather than the three ecomorphs. It is not necessary to make these changes, but I think the authors should justify the species sampling if possible, or note the limitations of their approach.

Experimental design

No apparent issues from my perspective.

Validity of the findings

No major issues, although the following points could be considered for revision.

L.322: “…there is no single living ecomorphology that converges on Hyaenodon in terms of the morphofunctional traits analyzed …”
I would suggest providing a more careful description. This study sampled only hyaenids and large felids among extant carnivores.

L.324
Another possible explanation may be a phylogenetic inertia. The Creodonta, including Hyaenodon, are a phylogenetic group separate from the Carnivora. The Creodonta and the Carnivora differ in many morphological features (e.g. the position of carnassial teeth in the dentition), which may account for the lack of a match in the morphofunctional traits.

L.339-352
I would suggest stating that pumas and jaguars (or all felids) represent meat specialist for readers unfamiliar with these characteristics.

L.354-372
In this paragraph, the results are explained by the ecological and behavioral traits of the spotted hyena, rather than bone crackers. This indicates a potential issue with using the ecomorph category labelled “bone cracker”, although no other living carnivore is likely to crack bones to the same extent as the spotted hyena.

Additional comments

The titles of Figs. 2, 3, and 4 are the same. I suggest distinguishing these figures by providing different titles for each.

I would suggest providing a description of Table S1.

---

## Round 0.2 · accepted · Accept

Congratulations again, and thank you for your submission.
Warm regards,